# Phase Diagrams of Polymerization-Induced Self-Assembly Are Largely Determined by Polymer Recombination

**DOI:** 10.3390/polym14235331

**Published:** 2022-12-06

**Authors:** Artem Petrov, Alexander V. Chertovich, Alexey A. Gavrilov

**Affiliations:** 1Faculty of Physics, Lomonosov Moscow State University, 119991 Moscow, Russia; 2Semenov Federal Research Center for Chemical Physics, 119991 Moscow, Russia

**Keywords:** block-copolymer micelles, polymerization-induced self-assembly, ATRP, computer simulations, dissipative particle dynamics

## Abstract

In the current work, atom transfer radical polymerization-induced self-assembly (ATRP PISA) phase diagrams were obtained by the means of dissipative particle dynamics simulations. A fast algorithm for determining the equilibrium morphology of block copolymer aggregates was developed. Our goal was to assess how the chemical nature of ATRP affects the self-assembly of diblock copolymers in the course of PISA. We discovered that the chain growth termination via recombination played a key role in determining the ATRP PISA phase diagrams. In particular, ATRP with turned off recombination yielded a PISA phase diagram very similar to that obtained for a simple ideal living polymerization process. However, an increase in the recombination probability led to a significant change of the phase diagram: the transition between cylindrical micelles and vesicles was strongly shifted, and a dependence of the aggregate morphology on the concentration was observed. We speculate that this effect occurred due to the simultaneous action of two factors: the triblock copolymer architecture of the terminated chains and the dispersity of the solvophobic blocks. We showed that these two factors affected the phase diagram weakly if they acted separately; however, their combination, which naturally occurs during ATRP, affected the ATRP PISA phase diagram strongly. We suggest that the recombination reaction is a key factor leading to the complexity of experimental PISA phase diagrams.

## 1. Introduction

Block copolymers are able to self-assemble in selective solvents; the typical morphologies observed in such systems include vesicles as well as spherical and cylindrical micelles. These aggregates can serve as nanoreactors and as imaging and drug delivery systems [1]. Block copolymer micelles and vesicles can be obtained by synthesizing block copolymers in a good solvent and adding a cosolvent that is poor for one of the blocks afterwards. This method of preparation of the block copolymer aggregates has a serious drawback: the addition of the cosolvent leads to a low concentration of the polymer product, typically less than 1% [2,3,4]. This problem is one of the key reasons for the limited implementation of block copolymer self-assembly in the industry [2,3,4].

One of the recently developed ways to overcome this challenge is to synthesize block copolymers directly in a selective solvent. Typically, the corona-forming block (A-block) is presynthesized and placed in a good solvent mixed with monomers for the core-forming block (B-block). The B-block monomers are chosen in such a way that they are soluble in the nonpolymerized state and nonsoluble when they form a polymer chain. Therefore, block copolymers appear in the selective solvent during the B-block synthesis, and thus are able to self-assemble without addition of cosolvents. Such a method of producing block copolymer aggregates is called dispersion polymerization-induced self-assembly (PISA) [3,5]. The concentration of the polymer product after PISA can reach 10–50%, a value that is an order of magnitude larger than that typical for the traditional method of cosolvent addition to presynthesized copolymers [2,5].

Usually, PISA is conducted using reversible deactivation radical polymerization (RDRP) techniques such as reversible addition–fragmentation chain-transfer (RAFT) [3,5,6], atom transfer radical polymerization (ATRP) [7,8,9], or nitroxide-mediated polymerization (NMP) [10,11,12]. Further, we focus on the case of ATRP PISA due to its popularity and the relative simplicity for its modeling compared to the RAFT PISA [13].

ATRP is more complex than living anionic polymerization from the standpoint of the chemical kinetics. First, ATRP includes the reaction of reversible activation–deactivation of the growing ends of polymer chains. Second, chain termination is present during ATRP in contrast to the anionic polymerization. At the same time, ATRP PISA phase diagrams feature significant complexity: they include multiple phase coexistence regions and exhibit a strong dependence of the aggregate morphology on the polymer concentration [8]. It is possible that the complexity of the ATRP kinetics induces the complexity of the ATRP PISA phase diagrams. First of all, termination via recombination leads to the formation of ABA triblock copolymers; since the polymer architecture may influence self-assembly [14,15], the presence of a termination via recombination may also affect the morphology of the ATRP PISA aggregates. Second, the rates of the reversible activation/deactivation and termination reactions relative to the rate of the propagation affect the polymer dispersity, and it is known that the dispersity can influence the process of the block copolymer’s self-assembly [16,17,18,19,20]. These considerations point out that it should be possible to control the morphology of ATRP PISA aggregates by controlling the kinetics of ATRP. However, the influence of different reactions occurring during ATRP on the ATRP PISA phase diagrams has been studied very poorly.

As it is very hard to discern between different factors affecting PISA phase diagrams in experiments, computer simulations can be employed to gain an insight into this complex problem. There exist several works on simulating the process of PISA [21,22,23,24,25]; however, those studies considered the simplest polymerization scheme that included only initiation and propagation reactions without chain activation–deactivation and termination reactions. In fact, the previous works modeled RDRP as an ideal living polymerization. As a result, the dispersity after PISA was significantly lower than typically observed in experiments [21,24]. Moreover, the absence of recombination in the previous models of polymerization led to the formation of only diblock copolymers during PISA. In those works, the influence of the presence of triblock copolymers on PISA phase diagrams was not studied. To the best of our knowledge, no attempt has been made to model PISA with a realistic polymerization scheme resembling the real process of RDRP.

In this work, we used the dissipative particle dynamics (DPD) technique [26,27,28,29,30] to simulate the process of ATRP PISA. In our model, we took into account the essential reactions occurring during ATRP. We assessed how the reversible activation–deactivation reaction and the reaction of termination via recombination affected the PISA phase diagrams. The former reaction led to an increase in the polymer dispersities up to 1.2; similar values are experimentally observed in well-controlled PISA [3,31]. However, despite this increase in the dispersity, ATRP PISA without the termination reaction yielded almost the same phase diagram as PISA modeled by an ideal living polymerization. On the other hand, termination affected the phase diagram strongly, shifting the transition between cylindrical micelles and vesicles and giving rise to a strong dependence of the aggregate morphology on concentration, which is typically observed in experiments. We believe that recombination is one of the key chemical processes affecting ATRP PISA.

## 2. Methods

To perform simulations, dissipative particle dynamics [26,27,28,29,30] (DPD) was utilized. DPD is a mesoscale molecular dynamics technique with a soft conservative force and explicit solvent. This method is well-suited for simulating polymer systems on a coarse-grained level. Macromolecules were modeled by the bead-and-spring model, with beads interacting by a conservative force (repulsion) $F_{\mathrm{ij}}^{c}$, a bond stretching force (only for connected beads) $F_{\mathrm{ij}}^{b}$, a dissipative force (friction) $F_{\mathrm{ij}}^{d}$, and a random force (heat generator) $F_{ij}^{r}$. The total force was given by: $F_{i}=\sum_{i\neq j}\left(F_{\mathrm{ij}}^{c}+F_{\mathrm{ij}}^{b}+F_{\mathrm{ij}}^{d}+F_{\mathrm{ij}}^{r}\right)$. The soft-core repulsion between the *i*th and the *j*th beads was equal to: (1)$$F_{\mathrm{ij}j}^{c}=\left\{\begin{array}{c} a_{\alpha \beta }\left(1-\frac{r_{\mathrm{ij}}}{R_{\mathrm{cut}}}\right)\frac{r_{\mathrm{ij}}}{r_{\mathrm{ij}}},r_{\mathrm{ij}}\leq R_{\mathrm{cut}},\\ 0,r_{\mathrm{ij}}>R_{\mathrm{cut}}. \end{array}\right.$$

Here, $r_{\mathrm{ij}}$ is the vector between the *i*th and the *j*th bead, $a_{\alpha \beta }$ is the repulsion parameter if the bead *i* has the type α and the bead *j* has the type β, and $R_{\mathrm{cut}}$ is the cutoff distance. $R_{\mathrm{cut}}$ is usually taken as the length scale, i.e., $R_{\mathrm{cut}}=1$[30]. In the present work, $a_{\alpha \alpha }=25$ was used ($a_{\alpha \alpha }$ represents the interaction parameters between alike beads, i.e., when β=α). In that case, the interaction parameters $a_{\alpha \beta }$ and a more common Flory–Huggins parameter χ are linearly related to each other [30]: $a_{\alpha \beta }=\chi /0.306+25$, α≠β. If two beads (*i* and *j*) are connected by a bond, there is also a simple spring force acting on them: $F_{\mathrm{ij}}^{b}=-K\left(r_{\mathrm{ij}}-l_{0}\right)r_{\mathrm{ij}}/r_{\mathrm{ij}}$, where *K* is the bond stiffness and $l_{0}$ is the equilibrium bond length. The following set of parameters was used to simulate bonds: K=4, $l_{0}=0$. A more detailed description and parameters discussion of the standard DPD scheme can be found elsewhere [26,27,28,29,30,32].

The ATRP reaction was modeled as described in ref. [32]. The algorithm is shown schematically in Figure 1. During polymerization, two beads could form a permanent bond when they were located closer than $R_{\mathrm{cut}}=1.0$ (i.e., when they were close enough to start interacting through the volume potential) from each other in space (the details of this procedure are given in refs. [21,32]). The reaction routine was run every $N_{\mathrm{stp}}=200$ DPD time steps. This number was large enough to ensure local equilibration of the system between consequent reaction routine runs, and small enough to result in a continuous reaction process. In the reaction routine, bonds could be formed with a certain probability depending on the type of reaction. Our systems contained three types of beads: A-type beads (solvophilic blocks, macroinitiators), B-type beads (solvophobic monomers and monomer units), and solvent (S). The initial system state was a homogeneous mixture of macroinitiators, solvent, and nonpolymerized B-type monomers. A single macroinitiator was formed by six A-type beads; the end of the chain $A^{*}$ could form a bond (Figure 1). The $A^{*}$ beads were in the dormant state with a probability $p_{A,\mathrm{dorm}}=0.99$; in this state, the $A^{*}$ beads were unable to form bonds (i.e., they were treated as nonreactive). When an $A^{*}$ bead was not in the dormant state, it was able to form a bond with a B-type bead with a probability $p_{i}=0.999$; this B-type bead started to form the B-block. B-type beads at the ends of growing B-blocks ($B^{*}$) were in the dormant state with a probability $p_{B,\mathrm{dorm}}=0.9$. In the active state, a B-type bead at the end of a B-block ($B^{*}$) could form a bond with a nonpolymerized B-type bead with a probability $p_{p}=0.05$. It is worth mentioning that in the present work, a generalized coarse-grained model of ATRP was utilized, and the transition metal catalyst was not simulated explicitly. Instead, its presence was taken into account implicitly in the activation/deactivation of the growing chains as a reversible change of the state (dormant/active) of the end-bead of the chains.

For simplicity, termination proceeded only via recombination: nondormant $A^{*}$ or $B^{*}$ ends of two polymer chains could form a bond with a probability $p_{t}$ if they were located closer than $R_{\mathrm{cut}}$ to each other. In this work, we adjusted $p_{t}$ to investigate the self-assembly close to ≈100% conversion for the following three cases: (i) no recombination ($p_{t}=0$), i.e. only diblock copolymers were formed in the system; (ii) recombination is present, and ≈50% of chains underwent recombination; and (iii) recombination is present, and ≈95% of all chains were terminated. Case (ii) can in general be considered as a simplified model of a more realistic ATRP PISA process, in which recombination via both recombination and disproportionation is present. For case (iii), the value of $p_{t}$ was fixed at 0.05, while for case (ii), it was adjusted to achieve the target fraction of terminated chains (≈50%) at a high conversion degree.

The values of the aforementioned reaction probabilities were chosen so that the molecular weight distributions (MWDs) (i) had experimentally reasonable dispersities for $p_{t}=0$ (no recombination) and (ii) were unimodal, i.e., had a single peak at all conversions (Figure 2). Typical dispersities achieved in the systems without recombination ($p_{t}=0$) were around 1.1–1.2 at 100% conversion (Figure 2 and Appendix A). When the reaction probabilities were adjusted in an attempt to further widen the distributions, the second peak started to appear in MWDs at high polymer volume fractions and low $N_{B}/N_{A}$ values.

The Flory–Huggins parameters χ of all interactions were equal to zero except for the interaction between the A-type beads and B-type beads ($\chi_{\mathrm{AB}}$) and between the solvent and B-type beads ($\chi_{\mathrm{SB}}$). We set $\chi_{\mathrm{AB}}=\chi_{\mathrm{SB}}=1.9$ as in ref. [21]: such choice ensured poor solvent conditions for the growing chains but did not lead to the precipitation of the nonpolymerized B-type beads. All bond potential and conservative force coefficients were set to make phantom chains for faster equilibration as described in ref. [21]. In our simulations, we used a cubic simulation box with periodic boundary conditions with a size of 80×80×80 DPD units. The total number of beads in one system was equal to $1.536\times 10^{6}$ (DPD density ρ=3); such a large size of the simulation box was chosen to reduce the influence of the finite-size effects. The integration timestep was equal to Δt=0.04.

To verify that the ATRP simulation algorithm described above modeled ATRP realistically, we analyzed the molecular weight distributions (MWDs) in the systems with a nonzero recombination probability at different polymer concentrations and compositions (Figure 2; see also Appendix A). The data in Figure 2 and Appendix A (black lines) show that MWDs were unimodal (as in the majority of experiments [8,9,33,34]). Moreover, our data suggest that the MWDs from our simulations changed with the polymer concentration and composition (Appendix A). In addition, we chose the reaction probabilities to speed up the calculations of ATRP; as a result, the ratios of the reaction probabilities did not correspond to the values typical for experimental systems. However, we additionally tested the reaction probabilities ratios closer to the experimental ones and did not observe any significant differences in the MWDs. Therefore, we believe that it is unlikely that our results were influenced by this choice of elevated reaction probabilities (see Appendix A).

After devising a computationally efficient way to simulate ATRP, we calculated the ATRP PISA phase diagrams. However, polydisperse polymers having different architectures due to recombination self-assemble into thermodynamically equilibrium structures very slowly. To overcome this challenge, we developed a fast algorithm of the phase diagram calculation (Appendix A). The fast algorithm consisted of the following three stages: (i) ATRP polymerization until 100% conversion was reached and the needed number of chains underwent recombination; (ii) precipitation of the chains into a single aggregate; and (iii) transformation of the aggregate into an equilibrium morphology. The phase diagrams for PISA without recombination (only diblock copolymers present in the system) were calculated both by the fast algorithm and by natural system evolution, and these two phase diagrams coincided (Appendix A). Given that the morphology equilibration essentially started from two very distinct initial states (i.e., precipitated and homogeneous) but the resulting morphologies were the same, we believe that the equilibrium phase diagrams were obtained. Moreover, this validation of the fast algorithm suggested that it could be used on its own to obtain equilibrium phase diagrams. Thus, the phase diagrams for ATRP PISA with recombination were calculated only via the fast algorithm, since it was very hard to obtain the equilibrium morphologies by natural evolution in a feasible time due to the very slow system relaxation (presumably because of the presence of triblock copolymers).

## 3. Results and Discussion

### 3.1. ATRP without Recombination

First, we studied the process of ATRP PISA without recombination ($p_{t}=0$). We obtained phase diagrams at 100% conversion, i.e., all B-type monomers were polymerized. The studied polymer volume fractions were 9%, 13%, 17%, and 21%. The phase diagram for this process is shown in Figure 3a.

In the phase diagram, three regions were observed: (i) a region of coexistence of spherical (sph) and cylindrical (cyl) micelles (similarly to our previous work [21]); (ii) pure cylindrical micelles; and (iii) vesicles (ves). We did not observe the regions of coexistence between the last two morphologies: there was a single transition line, almost independent of the concentration (Figure 3a). In the present work, we did not study the transition between the sph+cyl region and the region of pure spherical micelles that should occur at smaller values of $N_{B}/N_{A}$. We obtained the position of the sph+cyl<->cyl transition, and it was found to be almost the same as in [21]. Due to some ambiguity in the determination of the position of that transition, we mostly focused on the cylinders–vesicles transition (Figure 3b). To assess the influence of the ATRP-specific chemical reactions on the ATRP-induced self-assembly, we compared our data with two reference phase diagrams: (i) the self-assembly of monodisperse diblock copolymers and (ii) the self-assembly induced by living polymerization. Living polymerization was modeled by only two reactions: initiation and propagation, and no dormant chains were present. The reference phase diagrams were obtained in our previous work [21]. From Figure 3b, one can clearly see that the cylinders–vesicles transition line for ATRP PISA with no termination was shifted to higher values of B-block lengths compared to the two reference diagrams (Figure 3b). This phenomenon may be explained by an increase in the B-block dispersity: systems with more polydisperse B-blocks were shown to yield phase diagrams with a shifted line of transition between cylinders and vesicles [21]. Since the reversible activation–deactivation (i.e., the presence of dormant chains) led to an increase in the B-block dispersity compared to living polymerization in all systems (Appendix A), the aforementioned transition line was shifted to higher values of B-block lengths in the case of ATRP PISA. This shift was notable compared to the system of monodisperse diblock copolymers but was much less significant in comparison with the living PISA (Figure 3b). These data suggest that chemical reactions specific to ATRP do not have a strong influence on the PISA phase diagrams in the absence of recombination.

### 3.2. ATRP with Recombination

Next, we studied the self-assembly in the systems with nonzero recombination probabilities. We started with the case of ≈95% terminated chains; since only recombination was considered, those chains formed ABA triblock copolymers. The examples of molecular weight distributions for such systems are shown in Figure 2 and Appendix A. After reaching 100% conversion and 95% fraction of recombined chains, we stopped the reaction and studied the self-assembly behavior in the systems; the cyl<->ves transition points were tested in three independent runs. The phase diagram is shown in Figure 3a,b. The transition between cylinders and vesicles occurred at much larger $N_{B}/N_{A}$ values than in the systems with no termination. Moreover, we observed a pronounced dependency of aggregate morphology on concentration in the ≈95% terminated chains case.

To find out the reasons for such behavior, we carried out simulations of the self-assembly of monodisperse ABA triblock copolymers (Appendix A). To our surprise, we found that the transition between cylindrical micelles and vesicles occurred at very similar $N_{B}/N_{A}$ values as in the case of monodisperse diblock copolymers. Therefore, the architecture of monodisperse block copolymers alone did not affect the transition between cylindrical micelles and vesicles strongly.

Further, we compared the dispersities of the B-blocks in the systems with no recombination and with ≈95% terminated chains. We discovered that at Φ=21%, the recombination reaction affected the B-block dispersity weakly. For example, for the systems with $N_{B}/N_{A}\approx 6.3$, the B-block dispersity increased from 1.15 (no termination) to 1.18 (≈95% terminated chains). At Φ=13%, for the systems with $N_{B}/N_{A}\approx 6.3$, the B-block dispersity increased from 1.11 to 1.2 after increasing the fraction of terminated chains from 0% to ≈95%. This increase in the B-block dispersity was approximately equal to the difference of chain dispersities between the ideal living polymerization [21] and ATRP without termination (Appendix A); however, the two latter processes yielded very close phase diagrams (Figure 3b). Hence, the only explanation for the dramatic change of the phase diagram after taking into account the recombination reaction is that the B-block dispersity affects the self-assembly of ABA triblock and AB diblock copolymers somehow differently. In particular, we suggest that cylinders formed by ABA triblock copolymers are much more susceptible to changes in the B-block dispersity than those formed by AB diblock copolymers.

This fact also explains the strong concentration dependence of the cylinder–vesicle transition (Figure 3b). This dependence arises due to the slight changes of B-block dispersity, and not due to the change of polymer concentration itself; simulations described in Appendix A, substantiate this hypothesis.

Figure 4 provides a qualitative explanation of the phenomenon described above by comparing the packing of polymer chains in the cylindrical micelles. We suppose that there are two qualitatively different conformations of an ABA triblock copolymer in a micelle. To form a conformation of the first type, the two A-blocks reside in close proximity to each other in the corona of the micelle, and a B-block forms a “loop” close to the core-corona interface. The copolymer chains having the second type of conformations go through the micelle, having A-blocks on the opposite sides of the micelle. Obviously, the second type of conformations can be realized only by long enough chains. An increase in the B-block dispersity leads to a widening of the B-block length distribution, introducing more blocks with higher lengths, which are able to go through the micelle’s core center. We believe that this type of conformations stabilizes cylindrical (i.e., more curved) micelles, since it allows the system to reduce the entropy losses due to the redistribution of the blocks of different lengths in the micelle core, with longer blocks occupying the center. As a result, higher $N_{B}/N_{A}$ values are needed to form a vesicle compared to less polydisperse systems. There have been experimental evidences of such “separation” of the core-forming blocks of different lengths in the cores of diblock copolymer micelles [35]; however, owing to two A blocks which reside in the corona and the subsequent conformational peculiarities of the core-forming B-blocks, triblock copolymers seem to be more sensible to changes in the B-block dispersity compared to diblock copolymers.

To characterize the chain packing inside the micelles in a more quantitative manner, we compared two systems at a fixed value of $N_{B}/N_{A}\approx 6.67$ and polymer volume fraction of Φ=21%. One of these systems was obtained by the usual ATRP PISA approach with the fraction of terminated chains equal to 95% (in this system, the dispersity of the B-block of the ABA triblock copolymers was equal to 1.17), while the other was obtained by removing part of the solvent from the corresponding system at Φ=13% as described in Appendix A (in this system, the dispersity of the B-block of the ABA triblock copolymers was equal to 1.23). In the former system, vesicles were obtained (Figure 3b), while in the latter system, cylinders were observed (Appendix A). We compared the distributions of the end-to-end distances of the central B-blocks of the ABA triblock copolymers in these two systems; the chain ensemble forming the cylinder had more chains having larger end-to-end distances (Appendix A). This supported our hypothesis that the shape of the MWD influenced the micelles morphology through the chain packing.

The self-assembly behavior of systems with intermediate fraction (≈50%) of terminated chains corroborated the aforementioned arguments (Figure 3b). To prepare such systems, we chose $p_{t}=0.0215$ at Φ=21% and $p_{t}=0.015$ at Φ=13%; two different termination probabilities were chosen to avoid bimodal molecular weight distributions and to reach high conversions (≈98%). The reaction was stopped when 50% of the chains were terminated, and the self-assembly behavior was studied afterwards. Figure 3b demonstrates that the transition line between cylinders and vesicles for ≈50% terminated chains lies in between the transition lines for the systems with no termination and with ≈95% terminated chains. This is expected from our qualitative picture explaining the different effect the B-block dispersity has on the transition (Figure 4).

## 4. Conclusions

In our work, we developed a DPD-based model of ATRP PISA that reproduced the key elements of the ATRP chemical reaction. We demonstrated that the termination via recombination strongly affected ATRP PISA phase diagrams; presumably, this happened because of the enhanced influence of the B-block dispersity on the self-assembly of ABA triblock copolymers. Due to this effect, ATRP PISA with recombination yielded phase diagrams that had a strong dependence of the aggregate morphology on the polymer concentration similar to experiments.

Qualitatively, our findings are in agreement with existing experimental data. For example, in some studies on ATRP PISA [8,36] it was observed that wormlike micelles could turn into vesicles upon an increase in the polymer concentration at a fixed composition. One of the main results of our work agrees well with these data: the introduction of a high fraction of chains terminated via recombination leads to a tilted cyl–ves transition line. Namely, for certain $N_{B}/N_{A}$ values, we observed cylindrical micelles at Φ=13% and vesicles at Φ=21% (Figure 3b). At low fractions of chains terminated via recombination, the cyl–ves transition lines were almost vertical (i.e., there was no dependence of the morphology on the polymer concentration); therefore, we can speculate that recombination is one of the key factors controlling the micelles’ morphology in the real PISA process.

## Figures and Tables

**Figure 1 polymers-14-05331-f001:**
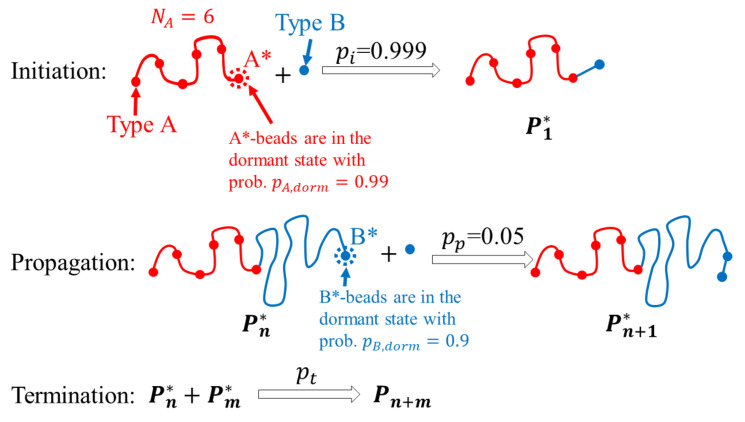
Algorithm of the ATRP simulation.

**Figure 2 polymers-14-05331-f002:**
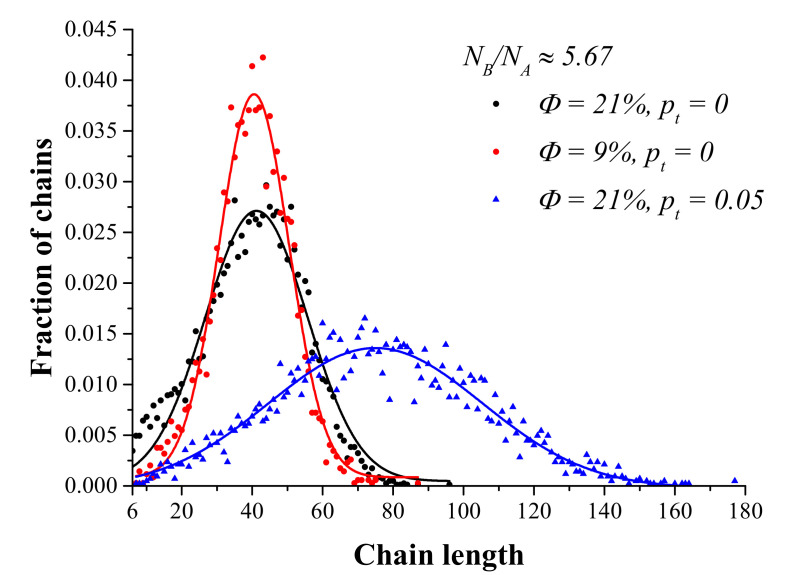
Molecular weight distributions of the modeled systems at a fixed chain composition ($N_{A}=6$, $N_{B}=34$) at different polymer volume fractions Φ and termination probabilities $p_{t}$ at 100% conversion; 95% of the chains in the system with $p_{t}=0.05$ underwent termination. Dispersities (values of $M_{w}/M_{n}$) are equal to 1.13, 1.07, and 1.14 for the black, red, and blue curves, respectively.

**Figure 3 polymers-14-05331-f003:**
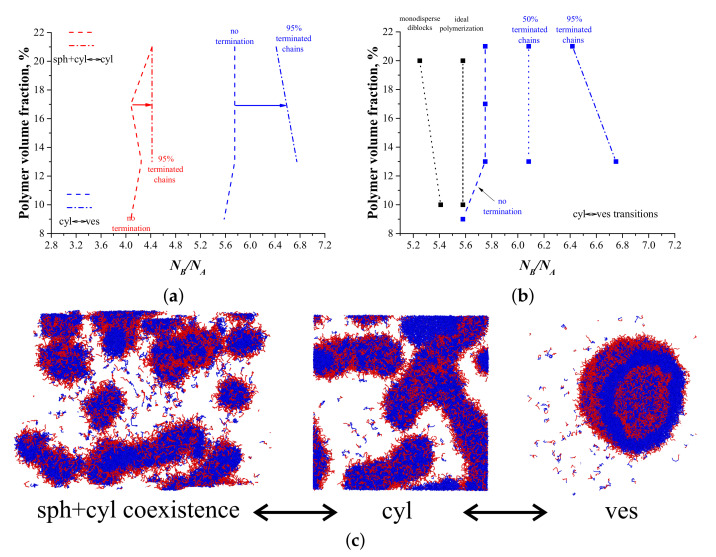
(**a**) Phase diagram for the systems without recombination ($p_{t}=0$, transition lines are dashed) and for the systems with the fraction of terminated chains via recombination equal to ≈95% ($p_{t}=0.05$, transition lines are dash-dotted). Sph+cyl and cyl regions are to the left and to the right side of the red lines, respectively. Cyl and ves regions are to the left and to the right side of the blue lines, respectively. (**b**) Transition lines between cylinders and vesicles for the systems with different fractions of terminated chains. Squares show the simulation data. Transition points for the reference systems (monodisperse diblock copolymers and ideal polymerization PISA at 10% and 20%) were obtained in ref. [21] and are shown in black. The cyl<->ves transition points were tested in three independent runs for the systems with 95% of terminated chains. (**c**) Snapshots of the observed morphologies.

**Figure 4 polymers-14-05331-f004:**
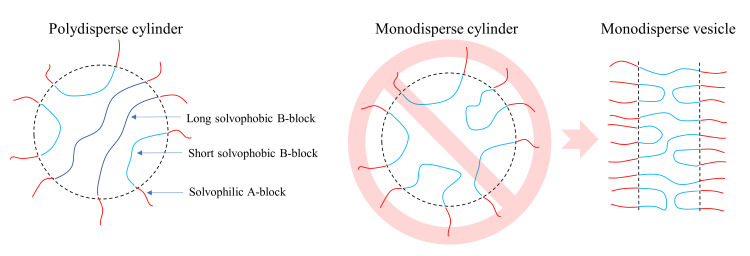
Schematic representation of the packing of ABA triblock copolymers inside micelles.

## Data Availability

The data that support the findings of this study are available from the corresponding author upon reasonable request.

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
