# Peer review of "Phase Diagrams of Polymerization-Induced Self-Assembly Are Largely Determined by Polymer Recombination"

_polymers, 2022, doi:10.3390/polym14235331_

Round 1
Reviewer 1 Report
In the paper, the authors used dissipative particle dynamics simulations to obtain atom transfer radical polymerization-induced self-assembly (ATRP PISA) phase diagrams and to determine the influence of the termination via recombination on ATRP PISA phase diagrams. They developed a fast algorithm for determining the equilibrium morphology of block copolymer aggregates. The authors assessed how the reversible activation-deactivation reaction and the reaction of termination via recombination affected the PISA phase diagrams.
The paper is a continuation of the author’s work previously published in Macormolecules (https://doi.org/10.1021/acs.macromol.7b00180, https://doi.org/10.1021/acs.macromol.1c02081) and Polymers (https://doi.org/10.3390/polym12112599) aimed at atom transfer radical polymerization-induced self-assembly.
The simulation was well-designed and the paper the article has the appropriate arrangement, however, several aspects need to be completed:
1) The ATRP process was simulated on the basis of the scheme from the paper https://doi.org/10.1021/acs.macromol.7b00180, and in turn, this article simulates the RDRP based on the diagram in the article published in 2006. Up to now, the ATRP has been extensively developed, especially low ppm ATRP techniques, therefore the authors should consider performing a simulation on the basis of newer ATRP-specific schemes and parameters, the latest information on the individual components of the rate constants, etc.
2) Usually, simulations should be an addition to or confirmation of previously conducted experiments, it should be a support for experimental data, which also allows the use of correct schemes and parameters for the simulation. The authors didn’t conduct experiments to confirm the correctness of the obtained data. Therefore, experiments should be conducted or the authors should indicate the model verification with the available literature data and good agreement between experimental and simulated data for both.
3) The authors should also significantly expand the Conclusions and include also a reference to the experiments carried out earlier and described in the literature.
Reviewer 2 Report
Please see the attachment

Round 2
Reviewer 1 Report
The authors addressed all the comments, accurately explained the ambiguities, and supplemented the missing information.
I suggest accepting in the present form.
Reviewer 2 Report
I thank the authors for modifying and completing the manuscript. In my opinion, the manuscript can now be published in its present form.